# Blackleg: A Review of the Agent and Management of the Disease in Brazil

**DOI:** 10.3390/ani14040638

**Published:** 2024-02-16

**Authors:** Ananda Iara de Jesus Sousa, Cleideanny Cancela Galvão, Prhiscylla Sadanã Pires, Felipe Masiero Salvarani

**Affiliations:** 1Instituto de Medicina Veterinária, Universidade Federal do Pará, Castanhal 68740-970, PA, Brazil; ananda.sousa@castanhal.ufpa.br; 2Centro de Desenvolvimento Tecnológico, Biotecnologia, Universidade Federal de Pelotas, Pelotas 96010-900, RS, Brazil; annymedvet@gmail.com; 3Ânima Educação, São Paulo 05435-001, SP, Brazil; prhiscylla.pires@animaeducacao.com.br

**Keywords:** *Clostridium chauvoei*, toxin–host interaction, virulence factors, myonecrosis, vaccination

## Abstract

**Simple Summary:**

Blackleg is a bacterial disease that primarily affects cattle but can also affect other cloven-hoofed animals such as sheep and goats. It is caused by the spore-forming bacterium *Clostridium chauvoei* and is prevalent in many parts of the world, including Brazil. Vaccination is one of the most effective ways to prevent blackleg. In Brazil, various types of vaccine are available, and they are typically administered to young animals. Brazil is a vast country with diverse climates and ecosystems. This can present challenges in terms of disease management strategies that are effective across all regions. In more remote areas, access to veterinary care and supplies may be limited, which can hinder timely diagnosis and treatment.

**Abstract:**

The genus *Clostridium* is an important group of pathogenic and nonpathogenic Gram-positive anaerobic bacteria with a sporulation capacity and wide distribution in different environments, including the gastrointestinal tracts of healthy and diseased animals and humans. Among the pathogenic species of the genus, *Clostridium chauvoei* stands out as a histotoxic agent. It causes significant myonecrosis such as blackleg, a disease with high lethality, especially in young cattle, and is responsible for significant livestock losses worldwide. The pathogenicity of the disease is complex and has not yet been fully elucidated. Current hypotheses cover processes from the initial absorption to the transport and deposition of the agent in the affected tissues. The virulence factors of *C. chauvoei* have been divided into somatic and flagellar antigens and soluble antigens/toxins, which are the main antigens used in vaccines against blackleg in Brazil and worldwide. This review provides important information on the first and current approaches to the agent *C. chauvoei* and its virulence factors as well as a compilation of data on Brazilian studies related to blackleg.

## 1. Introduction

The genus *Clostridium* comprises more than 200 species of bacteria, most of which are nonpathogenic and live in the environment, in plants, and in the skin and mucosa of healthy and/or diseased animals and humans (especially in the gastrointestinal tract). Some of these species contaminate and grow in foods of plant and animal origin, causing deterioration [1]. These are strictly anaerobic bacteria, but the degree of tolerance to the presence of oxygen is particular to each species. Clostridia are usually Gram-positive rods that, under adverse conditions, such as the absence of nutrients and the presence of oxygen, can take on a resistant morphology called spores [2].

Between 40 and 50 species are associated with clinical conditions in domestic animals and humans. Most pathogenic species, approximately 30, are considered secondary pathogens. Only approximately 15 of them are thought to be important primary pathogens [1,3]. The main pathogenic species cause disease through the production of exotoxins, which are responsible for the development of lesions and clinical signs. These are among the most potent toxins of microbial origin and are considered the basis for identifying and differentiating the various pathogenic species of *Clostridium*. Clostridia are responsible for the production of 20% of the main and most potent toxins of bacteriological origin [3].

Clostridial toxins comprise a wide variety of proteins with different sizes, structures, and mechanisms of action. They also differ in their ability to diffuse in the host organism and the site of action. Some act locally, while others enter the bloodstream, spreading to different organs and tissues. However, in some clostridioses, the toxins and the multiplication of the agent are equally important in the pathology, such as in myonecrosis [4,5].

Many clostridia cause diseases of public health importance through the action of their toxins. These proteins trigger specific clinical signs, which are related to certain types of clostridiosis. Notable examples include the pseudomembranous colitis of *C. difficile*, the spastic paralysis of infection by *C. tetani*, and the flaccid paralysis of *C. botulinum* [6]. Other clostridia, although producing different toxins, cause diseases with similar clinical signs. Myonecrosis, for example, can be caused by the following species alone or together: *C. septicum*, *C. chauvoei*, *C. novyi* type A, *C. perfringens* type A, and *C. sordellii*, but the lesions are similar regardless of the etiology of the disease. Despite the clinical and pathological similarities, each Clostridium species has its own mechanism that culminates in the observed changes. Thus, few generalizations about the mechanisms of virulence can be made, requiring the study and individualized understanding of the various species [7].

The present article aims to review the main virulence factors and prevention of *C. chauvoei* infection and provide a compilation of data on Brazilian studies on blackleg.

## 2. Myonecrosis

Histotoxic clostridia are pathogens common to humans and animals that can cause two forms of necrotizing myositis, usually fatal, called gas gangrene and blackleg. Gas gangrene or malignant edema is an exogenous infection that affects practically all species of veterinary interest, such as cattle [8,9], sheep, goats [10], pigs [11], horses [12], dogs [13], and birds [14]. Caused by one or more of *C. septicum*, *C. chauvoei*, *C. novyi* type A, *C. perfringens* type A, and *C. sordellii*, gas gangrene requires, for example, the contamination with these agents in wounds resulting from surgical practices without aseptic care [2,15].

Unlike gas gangrene, blackleg is an endogenous infection that affects domestic and wild ruminants, especially cattle aged 4–6 weeks, and results from the activation of latent *C. chauvoei* spores present in the muscle [7,16,17]. This is considered an exclusively extracellular bacterium [16]. The pathogenesis of this disease is not well understood. The main hypothesis is that this agent is absorbed from the intestinal mucosa and deposited in the muscle after transport via the blood circulation by tissue macrophages [16,18]. To test this hypothesis, Pires et al. [15] evaluated the survival of vegetative cells and spores of *C. chauvoei* after phagocytosis by a murine and bovine macrophage cell line and demonstrated that both remain viable after internalization by these phagocytic cells. In summary, macrophages are integral to the immune response against *Clostridium chauvoei* in blackleg disease. While they play a critical role in attempting to eliminate the bacteria and their toxins, their actions can contribute to both the resolution of the infection and the development of tissue damage due to the inflammatory response. Understanding the interplay between macrophages, the immune response, and bacteria is crucial for developing strategies to modulate the immune system effectively, potentially reducing the severity of blackleg and improving treatment outcomes.

The factors responsible for the germination of spores in the muscle have not yet been fully established. It is believed that traumas, especially in large muscle masses, create an environment conducive to germination and the consequent production of toxins, which usually culminate in rapid muscle necrosis and the death of affected animals [18,19].

## 3. *Clostridium chauvoei*

Among the pathogenic microorganisms of the genus Clostridium, *C. chauvoei* is a particularly important microorganism in veterinary medicine. The etiological agent of both blackleg and gas gangrene, this bacterium has been responsible for considerable economic losses to livestock worldwide [16,20]. Although gas gangrene and blackleg were described in the mid-19th century, *C. chauvoei* was only described in 1887 [21]. The identification of the agent was delayed due to the great complexity of its bacteriological isolation, an obstacle that holds today. Several factors are involved in this diagnostic difficulty, including the specific growth requirements of the bacterium and the fastidious growth of this pathogen [22,23]. In addition, clinical samples are often contaminated with other anaerobic bacteria that spread after the death of the animal [24]. The contaminating microorganisms are less demanding and grow faster than *C. chauvoei*, inhibiting its growth [25].

Like most clostridia, *C. chauvoei* is a Gram-positive, motile rod measuring 0.6 × 3–8 μm, occurring singly or in pairs [20]. The bacterial spores are oval, central to the subterminal, causing the deformation of the mother cell. Cultivation on solid media allows the production of colonies with different morphotypes, including colonies regular with low convex, whitish-grey, large (5 mm in diameter), but may also form a continuous “carpet” of bacteria, (the bacteria are swarming) or as apleomorphic in structure. Identification can be facilitated by culturing on sheep blood agar, as it results in colonies surrounded by extensive hemolysis [18,22].

Thus, differentiating *C. chauvoei* from other clostridial species via culture is extremely difficult because of the various forms of growth; carbohydrate fermentation is nonspecific. Therefore, it is necessary to use immunological diagnostic methods [25], such as direct immunofluorescence [26,27] and PCR [13,28], to identify *C. chauvoei*.

*C. chauvoei* is a widespread microorganism, especially in cattle production areas such as pastures and confinement pens [29,30]. This agent contaminates soil, pastures, drinking troughs, and pens via the decomposition of carcasses and the excretion of spores in feces. Once contaminated, the area can remain contaminated for years [25,31].

The maintenance of viable spores in the environment, which can return to a vegetative state if they detect ideal conditions for germination, is the most significant factor in the infection of animals. The primary transmission route involves ingesting spores present in contaminated feed, pasture, and water, as direct animal-to-animal transmission does not occur. Environmental contamination allows the perpetuation of the agent, especially in farms where the disease is endemic [17,32,33]. Animals infected with *C. chauvoei* may exhibit loss of appetite, high fever, myonecrosis, an increase in the volume of affected muscles, and lameness [34,35]. The rapid course of the disease makes the early identification and treatment of these clinical signs difficult, indicating that vaccination is the best alternative for preventing blackleg.

Despite its recognized importance as a pathogenic agent in domestic animals, especially cattle, few studies have been conducted to better characterize the agent and its mechanisms of virulence, as Ziech et al. [17] carried out in the Figure 1 below.

## 4. Virulence Factors

Moussa et al. [36] divided the *C. chauvoei* antigens into two groups: cellular antigens and soluble antigens. Cellular antigens are classified into somatic and flagellar antigens, while soluble antigens encompass mainly toxins. Soluble antigens are directly involved in the pathogenesis of blackleg [17].

### 4.1. Soluble Antigens

Until recently, it was assumed that this clostridium produces at least five toxins: an oxygen-stable hemolysin (alpha), a DNase (beta), a hyaluronidase (gamma), an oxygen-sensitive hemolysin (delta) [19], and a neuroaminidase [37]. However, Frey et al. [38] have described a new toxin named CctA that is considered essential for the occurrence of the disease and the adequate immunization of animals [17,39].

#### 4.1.1. Alpha Toxin

The alpha toxin was first studied by Moussa et al. [36] and Verpoort et al. [40], who reported some of its activities and biological characteristics in nonpurified *C. chauvoei* culture filtrates. This exotoxin is a 27 kDa hemolysin that was later characterized as a necrotizing, hemolytic, and lethal protein [41].

The alpha toxin was first purified and partially characterized by Tamura et al. [42]. This research group demonstrated that the production of this toxin peaks in the logarithmic phase of bacterial growth and that the erythrocytes of sheep, cattle, and birds (chickens) are susceptible to the hemolytic action of this protein, while the erythrocytes of goats, rabbits, and guinea pigs were classified as partially resistant and those of horses as resistant. Hang’ombe et al. [41] also analyzed the sensitivity of erythrocytes in different animal species and found the same results as Tamura et al. [42], the only difference being in the sensitivity of sheep and bovine erythrocytes. The variation in sensitivity between species may be explained by the existence or absence of membrane receptors necessary for binding the toxin [41]. These same authors also found that the higher the temperature, the lower the amount of the toxin required to achieve the hemolysis of 50% of erythrocytes, indicating that the action of the alpha toxin is dependent on temperature. However, to date, no study has fully elucidated the mechanism of action of the *C. chauvoei* alpha toxin [41]. As there have been few conclusive studies on the toxins of *C. chauvoei*, the alpha toxin was long considered the main toxic factor of this species of Clostridium [19].

#### 4.1.2. Beta Toxin

The beta toxin, or DNase, is a deoxyribonuclease-type enzyme [43] found in more than 80% of *C. chauvoei* strains [44]. It is a 45 kDa thermostable protein responsible for nuclear degradation in muscle cells [19] and actively participates in cases of clostridial myonecrosis [45]. Although there is no correlation between the production of the alpha and beta toxins [31], both are considered fundamental in the process of gangrenous myositis triggered by this microorganism [44]. A previous study analyzed the complete sequences of 20 *C. chauvoei* strains isolated from different continents for 64 years and found the presence and conservation of two genes that are likely involved in DNase activity [17,46].

#### 4.1.3. Neuraminidase/Sialidase

Hyaluronidase is a general term for enzymes that can digest hyaluranate, present in hyaluronic acid and hyaluronan. Hyaluranate is a linear polymer of nonsulfated glycosaminoglycan found in many tissues and body fluids of higher organisms and is a major constituent of soft connective tissue, muscle, and skin. Hence, Gram-positive bacteria capable of producing hyaluronidase can initiate infections in the mucosa, subcutaneous tissue, and/or skin [47]. The genome of *C. chauvoei* has two different hyaluronidase genes: *nagH* and *nagJ* [17].

In the pathology of blackleg and gas gangrene, although the gamma toxin, a hyaluronidase, is not considered a lethal toxin, it is believed to be responsible for the significant disorganization of muscle tissue, with the loss of almost all structures [43]. During an infection of any type, the connective tissues and the skin provide a defense mechanism against infectious agents, resisting the penetration of these pathogens. However, hyaluronidase-producing bacteria, such as *C. chauvoei*, can weaken the restrictions imposed by the constitution of connective tissues, destroying them, and facilitating the propagation of the agents and toxins. In addition, the degradation products of hyaluronidase are disaccharides, which can be a nutrient source for pathogens [47].

#### 4.1.4. Delta Toxin

The delta toxin is a thiol-activated cytolysin [48]. It is sensitive to the presence of oxygen, which considerably reduces its hemolytic action. There is no in-depth characterization of the delta toxin. It is only known that it has similar mechanisms of action to the perfringolysin O produced by *C. perfringens* and tetanospasmin produced by *C. tetani* [19]. Thiol-activated cytolysins are so-called because they mainly act on cell membranes with a high cholesterol content. After anchoring onto cell membranes, these toxins oligomerize, forming pores [49].

#### 4.1.5. Neuraminidase

Neuraminidase, also called sialidase (*NanA*), belongs to a class of glycosyl hydrolases that release the terminal *N*-acetylneuraminic or sialic acid residues of glycoproteins, glycolipids, and polysaccharides. Neuroaminidase-type toxins have been detected in various microorganisms, such as viruses, bacteria, and protozoa [37]. Likewise, *C. chauvoei* is known to produce one neuraminidase that plays a significant role in the pathogenesis of the infection [50].

Neuraminidases are responsible for the cleavage of sialic acids in infected tissues and the destruction of erythrocytes, facilitating the propagation of the pathogen and the disease. Active sialidases comprise three domains: the N-terminal portion, which appears to be responsible for binding to its receptor; the central portion, which binds sialic acid; and the C-terminal enzymatic portion. It is a toxin of chromosomal origin encoded by the *nanA* gene, which is found in a wide variety of strains of *C. chauvoei*, including the reference sample ATCC 10092. The *nanA* gene determines the protein’s expression as a 150 kDa dimer, which results in a metabolically active protein of 72 kDa via proteolytic cleavage, presenting 82% similarity with the sialidase produced by *C. septicum* [51].

The sialidase isolated from *C. chauvoei* is a highly stable dimeric protein of 150 kDa. It showed no loss of activity after undergoing five cycles of freezing and thawing. Its activity was maintained at 37 °C in buffer C for at least 1 h [50]. This toxin remains active over a wide range of temperatures (4–50 °C) and pH values (4–7.5), with an optimum at 37 °C [33]. The main substrates of *C. chauvoei* sialidase are glycoproteins, which exist in large amounts in muscle tissue and erythrocytes, contributing to the hydrolysis of these cells and tissues. This suggests the toxin remains active even under unfavorable conditions and can support the growth and spread of the bacteria to cause muscle damage [50].

#### 4.1.6. Toxin A (CctA) of *C. chauvoei*

Frey et al. [38] described and initially characterized a new toxin produced by *C. chauvoei*: toxin A (CctA), a 33.2 kDa protein belonging to the beta-barrel family of pore-forming toxins within the leukocidin superfamily. The sequence analysis of the CctA gene showed significant similarity with the alpha-hemolysin genes of *C. botulinum* (50% identity and 80% amino acid similarity) and with the NetB of *C. perfringens* (44% identity and 60% amino acid similarity) and the *C. perfringens* beta toxin (33% identity and 51% amino acid similarity).

Approximately one-third of clostridial toxins, and many other bacterial toxins, are classified as pore-forming [3]. Beta-barrel pore-forming proteins are characterized by a conserved structure including domains that bind to specific receptors on target cells, followed by the internalization of part of the hydrophobic amino acid sequence in the host cell membrane that allows the anchoring of the protein. After binding and anchoring the protein, oligomerization occurs, usually into heptamers that form pores, which culminates in a large influx of extracellular content and the consequent lysis of the affected cells [52].

The gene encoding CctA is conserved, having been identified in strains found in different continents as well as in the reference strain ATCC (10092). Although the toxins described above were believed to be responsible for the symptoms of blackleg and gas gangrene, specific antibodies against CctA protected 90% of challenged animals, neutralizing all the cytotoxic and hemolytic effects promoted by the supernatant of *C. chauvoei* cultures [38]. These results are promising because this is the first time that vaccination with an exotoxin produced by *C. chauvoei* has protected such a high number of challenged animals.

### 4.2. Cellular Antigens

Cellular antigens of *C. chauvoei* can be subdivided into somatic antigens and flagellar antigens. Somatic antigens are related to bacterial cells and include agglutinogenic antigens, heat-stable O antigens, non-agglutinogenic antigens, and heat-labile antigens, which do not present the O and HO antigens [17,53]. Flagellar (H) antigens are thermolabile, and two distinct ones have been identified. Analyses of these antigens in different strains of *C. chauvoei* revealed that the somatic antigens are common to all of them, while one of the flagellar antigens differs between samples isolated from cattle and sheep [17,54]. Somatic antigens are essential immunogens linked to protection against *C. chauvoei* and are present in past and present vaccine formulations [17,55].

#### Flagellar Antigens

Bacteria move along chemical gradients using flagella, one of the smallest and most complex motors in the biosphere. This structure is practically identical in Gram-negative and Gram-positive bacteria. Flagella also aids in sensory function, allowing the bacterium to respond to chemical stimuli and avoid unfavorable environments, such as extreme pH values and high salt concentrations [56].

Flagella consist of a long helical filament composed of flagellin polymers that emerges from a “hook” connected to a basal body anchored inside the cell membrane. There is only one study on a basal model of the structure of the flagellum of *C. chauvoei*, which, according to Hamilton and Chandler [57], looks like that of other Gram-positive bacteria. Regarding the amino acid composition, the flagellin of *C. chauvoei*, which is encoded by the *fliC* gene, is similar to the flagellin of *Salmonella typhimurium*, and cysteine and tryptophan are not detected in its constitution. It has low percentages of proline, methionine, tyrosine, and histidine [58].

Bacterial flagella are best known for their role in bacterial motility. However, in certain pathogenic bacteria, these structures are also important in parasite–host interactions. Flagellated strains of *C. chauvoei* are more virulent than naturally non-flagellated strains [59]. Despite their constitutional similarity, the flagella of *S. typhimurium* are more involved in the invasive capacity of the strain than in its virulence, a fact that limits further comparisons.

While the structure of and interaction between the *C. chauvoei* flagellum and host cells have not been well characterized, the immunogenic potential of this structure has been widely analyzed. In contrast with the other clostridia, in which immunity against toxins has always been considered the predominant form of protection, immunity against *C. chauvoei* was considered for many years to be exclusively antibacterial [60]. The immunogenicity of the flagellum of *C. chauvoei* has been the subject of several studies. Techniques ranging from protein purification [61] to gene recombination [62] have been tested, but unfortunately, the results are controversial. Over the years, the only consensus regarding this protein has been that it is an important mechanism of virulence and immunogenicity; however, the manner and degree cannot yet be specified.

## 5. Prevention, Control, and Eradication

Vaccination against several clostridial pathogens, including *C. chauvoei*, has been used as a prophylactic measure worldwide for more than 70 years [63]. In particular, vaccination against *C. chauvoei* is a primary sanitary management measure. Scientific evidence of the effectiveness of vaccination and the antigens adopted in preventing the disease and deaths caused by the pathogen is still scarce [16]. Several antigenic compositions have been proposed. The first studies on immunogens sought to establish whether the best vaccine formulation should be based on bacterins, toxoids, or both. Due to the difficulties in cultivation, it was initially proposed that immunity was conferred only by the bacterins that composed the commercial antigens [55,64]. Efficient toxoids were only obtained in laboratory formulations, with controlled production conditions associated with the concentration of the toxins before the inactivation step [65]. Vaccine formulations were also evaluated in sheep by Coackley and Weston [66]. These researchers immunized three groups of sheep with toxoids, bacterins, or bacterins + toxoids. Only sheep vaccinated with toxoids died after challenge, which reinforced the initial idea that immunogens should be based on bacterins. The authors proposed that vaccinated animals would be protected for at least 18 months after vaccination.

As the bacterium has been considered one of the most important antigenic components, the authors of [67] sought to determine the most antigenic portion of the bacterial cell—the cell wall or the flagellum. These vaccine antibodies against cell wall of the flagellum of *C. chauvoei* are important in the opsonization of pathogens, increasing the efficiency of phagocytosis and protecting rats from experimental infection with *C. chauvoei* [68]. These authors [67,68] concluded that both could generate a protective response.

With the discovery of CctA [38], new perspectives in the study of prophylactic methods for blackleg have emerged. Researchers have performed the potency test recommended by the European Pharmacopeia using a CctA recombinant toxin and observed that all vaccinated animals survived the challenge. Thus, the new toxin is expected to confer protective immunity to animals, and the recombinant protein technique can facilitate the production of the immunogen.

## 6. Blackleg in Brazil

The occurrence of blackleg in Brazil dates to the 20th century. In 1905, the Ministry of Justice instructed the Oswaldo Cruz Institute to discover an effective way to minimize the losses caused by the disease, which had decimated the herds in São Paulo and Minas Gerais. In 1906, Alcides Godoy, a scientist at the institute, discovered the first veterinary vaccine in the country for the prophylaxis of infectious/contagious diseases: the vaccine against blackleg, better known at the time as Peste da Manqueira [69]. The immunogen produced was patented, and its patent was renewed after 15 years for another 15 years. The rights derived from the sale of the “Manguinhos” anti-southern disease vaccine were assigned to the institute on the condition that the respective income would be applied to the scientific activity of the institute. The disease was such a bottleneck for Brazilian livestock at that time that the sale of this vaccine was one of the institution’s main sources of income, enabling studies on Chagas disease, yellow fever, and leishmaniasis [70].

With the advent of vaccination, the disease was brought under control, so it was no longer the object of interest of Brazilian researchers. However, great interest remains among ranchers in protecting their herds by vaccinating them regularly and among public agencies in ensuring the quality of these immunogens. As such, clostridial vaccines are among the most sold vaccines in Brazil [2]. Despite the significant reduction in the incidence of this disease, outbreaks and isolated cases associated with the non-vaccination of animals are still described [28,71], and there are few reports of successful treatment [8].

In Brazil, there are at least three strands of study regarding blackleg: those related to the production of vaccines and vaccination protocol in animals, methods for evaluating antibody production, and diagnostic methods for the disease.

## 7. Production of Vaccines and Vaccination Protocols for Animals against Blackleg

Despite the massive immunization of Brazilian herds, the occurrence of blackleg remains an obstacle for Brazilian livestock. According to Assis-Brasil et al. [72], blackleg is the second most common disease detected in calves aged between four and six months. This has led some researchers to question the efficiency of the strain used as a reference in immunization. Previous studies indicated a difference in the protective immunity conferred by vaccinating animals with different strains against challenge with heterologous samples [8,60].

In light of these findings and the continued occurrence of blackleg in vaccinated herds, studies have conducted comparisons between the MT vaccine and local samples. Results from guinea pigs vaccinated with commercial products and subsequently exposed to the MT strain revealed that 95.46% of the tested vaccines failed to provide protective immunity. On the other hand, when a local strain was used instead of the vaccine sample, only 36.36% of the vaccines were approved [73]. Similar results were reported by Araujo et al. [74]. There is no consensus in the national or international literature on this subject. Other researchers found no difference in the immunity conferred by different strains of *C. chauvoei*. According to these authors, any vaccine failures have causes unrelated to the efficiency of the vaccines [75].

Despite the lack of consensus on the ideal composition of immunogens against blackleg, vaccination against *C. chauvoei* is part of the health management of Brazilian livestock. It is recommended that cattle be immunized between three and six months of age, with a booster every 30 days. After that, vaccination should be annual [76]. In places of high prevalence, the vaccination interval should be reduced to nine months [77].

On this topic, some studies were conducted in Brazil to establish the importance of passive immunity and the best protocol for vaccinating calves. According to Araújo et al. [78], passive immunity can be detected in calves up to three months after receiving colostrum from cows vaccinated up to 30 days before calving. The authors also emphasize the positive correlation between the antibody titer observed in the calves and the time of vaccination of the cow. Thus, the closer to calving the cows received the immunization, the higher the antibody titer of the calves at three months of age.

Regarding active immunity, Araújo et al. [74] evaluated the serological response of calves subjected to three different vaccination protocols against blackleg: vaccination at four and eight months of age; vaccination at eight months with a booster 30 days later; and receiving a single dose of the vaccine at eight months of age. Using enzyme-linked immunosorbent assay, the authors concluded that only the last vaccination schedule would not guarantee satisfactory antibody titers for the vaccinated animals when they were 10 months old.

CctA is one of the toxins produced by *Clostridium chauvoei*, and it belongs to a family of toxins known as beta-pore-forming toxins. Recent studies have indeed explored the potential of using CctA as a component for immunization against blackleg disease. The idea behind this research is to utilize the toxin in a controlled manner to trigger an immune response in animals without causing the actual disease. By doing so, the animal’s immune system can recognize and develop defenses against CctA, thereby providing protection against the bacterium that produces this toxin. These studies aim to develop vaccines that are effective, safe, and capable of providing robust immunity against Clostridium chauvoei infection without causing harm to the animals. However, while the research shows promise, it is important to note that vaccine development and testing involve various stages, including rigorous trials to ensure safety and efficacy before potential implementation in veterinary practices [79,80].

## 8. Diagnosis

In Brazil, despite the significant number of foci, most diagnoses are based only on inconclusive clinical signs and/or lesions at necropsy, with few reports of laboratory confirmation. The etiological diagnosis of blackleg and gas gangrene caused by *C. chauvoei* should be carefully addressed by veterinarians and laboratory technicians, given that this microorganism is a frequent invader of carcasses [2].

Bacterial isolation usually requires time-consuming and expensive laboratory procedures and trained personnel [22]. Direct immunofluorescence, the gold-standard diagnostic method, has been used worldwide and is considered a fast and safe technique. It requires a special microscope, primary antibodies labeled with fluorochromes, and fresh or specially processed material [27].

An alternative for diagnosing this disease is immunohistochemistry, which, by combining histological, immunological, and biochemical techniques, allows the localization of tissue components in situ by labeling with specific antibodies and marker molecules. The advantage of this technique is the use of formalin-fixed materials, allowing for long times between collection and laboratory processing without interfering with the diagnosis [79]. These long possible storage periods for any clinical material are highly relevant in veterinary medicine because most rural properties are far from diagnostic and research centers. In addition, the fixation of the tissues prevents their autolysis, preventing saprophytic clostridia from multiplying. All these factors increase the accuracy of the etiological diagnosis, information that allows the adoption of appropriate immunization schedules, in addition to serving as a reference for the vaccine production industry [7,80,81].

Polymerase chain reaction stands out as a viable technique for identifying microorganisms because it is extremely versatile, being able to identify the agents in cultures, fresh clinical specimens, or formalin-fixed and paraffin-embedded tissues [13,82]. It has the disadvantage of identifying the agent without colocation with the wound.

## 9. Conclusions

*Clostridium chauvoei* is the etiological agent of blackleg, a myonecrosis highly important in veterinary medicine due to heavy livestock losses worldwide stemming from its high lethality in ruminants, especially cattle. The acute course of the disease and its pathogenesis are not yet fully understood, making vaccination the main approach for control and prophylaxis used in Brazil and worldwide. Although the main virulence factors of *C. chauvoei* have been evaluated and described, studies are still needed to improve the characterization of the parasite/host relationships of these antigens so they can be applied to the development of new vaccine formulations, which we hope will increase the protection of *C. chauvoei*-immunized animals via simplified production strategies and lower costs compared with conventional vaccines.

## Figures and Tables

**Figure 1 animals-14-00638-f001:**
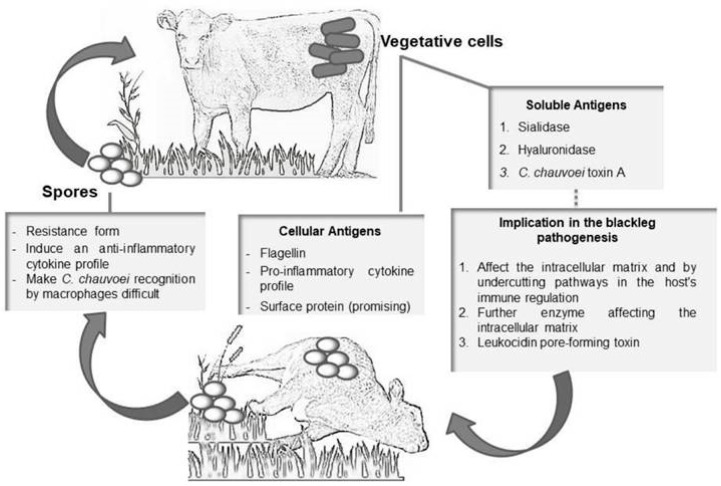
Schematic illustration of blackleg pathogenesis involving currently considered major virulence factors [17].

## Data Availability

Not applicable.

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
