# Peer review of "Blackleg: A Review of the Agent and Management of the Disease in Brazil"

_animals, 2024, doi:10.3390/ani14040638_

Round 1

Reviewer 1 Report

Comments and Suggestions for Authors

The authors have conducted a review of blackleg, focusing on the agent and management of the disease. It is always very interesting to find a review where the authors summarize all the information available on a topic. In this case, the authors take into consideration all the necessary aspects that are related to Clostridium chauvoei such as virulence factors, prevention and control, and vaccine production and vaccination protocols. The manuscript is very complete, interesting, and well written.

The only observation that I have are the following:

-       Have the authors considered changing the title? I would write a more general title, because as I have mentioned before, the authors are discussing general information and characteristics of this pathogen.

-       Change line 12: Blackleg caused by Clostridium chauvoei it is a highly contagious bacterial disease….

Author Response

Dear Reviewer 1,
Thank you very much for recognizing our effort in writing this review and for understanding the importance of this review from a scientific point of view.

Regarding the title, we are open to suggestions! Can you contribute your title suggestion to us? Could it be "Blackleg in Brazil: a review of the agent, virulence factors and prevention of the disease"? Or "Blackleg review"?

Change made to line 12: Blackleg caused by Clostridium chauvoei it is a bacterial disease...

Sincerely,
Felipe Masiero Salvarani

Reviewer 2 Report

Comments and Suggestions for Authors

This study gave a review on the agent and management of blackleg in Brazil, which is of interests. However, more work have to be done to improve the quarlity.

Major concern:

1. The review lacks of logic, which is hard to read. The authors can present the review in a more logic style, such as:

Blackleg

Pathogen

Pathogenicity

Diagnosis

Prevalence

Prevention and control 

2. The review can be visualized by using suitable Figures and Tables to summary main points   

Minor concern:

1. There are many spelling and formatting problems, e.g.

Line 12: delete "It", change "for" into "by"

Line 20: "Clostridium" should be italic

Line 22: change "tract" into "tracts"

Line 28:  "C. " should be italic

Line 36: "Clostridium" should be italic, revised elsewhere needed

Line 40: change "although" into "but"

Line 77: delete double blanks after "birds"

2. The formate of reference list needs to be revised, e.g., some references have doi no., but not for all.

Comments on the Quality of English Language

Extensive editing of English language required

Author Response

Dear Reviewer 2,
the article was evaluated by 4 reviewers and below I transcribe their speeches to be able to refute their comment "The review lacks of logic, which is difficult to read", perhaps the quality of the English could have caused this impression. But look at what feedback the article received:

"The authors have conducted a review of blackleg, focusing on the agent and management of the disease. It is always very interesting to find a review where the authors summarize all the information available on a topic. In this case, the authors take into consideration all the necessary aspects that are related to Clostridium chauvoei such as virulence factors, prevention and control, and vaccine production and vaccination protocols. The manuscript is very complete, interesting, and well written." Reviewer 1

"The review provides an in-depth examination of Clostridium chauvoei, a bacterium associated with blackleg in cattle and gas gangrene in various animals, highlighting its impact on veterinary medicine. The genus Clostridium encompasses numerous bacterial species, with C. chauvoei identified as a major pathogen causing significant economic losses globally. The article explores the bacterium's characteristics, clinical manifestations, and the main virulence factors contributing to diseases like blackleg. These factors include various toxins such as alpha toxin, beta toxin (DNase), hyaluronidase, delta toxin, neuraminidase, and a recently identified toxin, CctA. The study also highlights the historical context of blackleg vaccination in Brazil and underscores the ongoing challenges despite widespread immunization efforts. It discusses the diversity of vaccination protocols and the evolving understanding of immune responses, emphasizing the importance of continuous research for optimizing vaccine formulations. The article concludes by emphasizing the significance of accurate diagnostic methods, including direct immunofluorescence, immunohistochemistry, and polymerase chain reaction, in identifying C. chauvoei infections. The article is well-written, fluid, and highly comprehensive concerning the characterization of C. chauvoei."  Reviewer 3

Another point is your question that "The review can be visualized by using suitable Figures and Tables to summarize main points". Our objective was to do a real review without photos and tables, so that everyone could read the information. Several studies in the area of education and teaching demonstrate that review articles with photos and tables are not as effective in disseminating knowledge, since when summarizing information in a table or figure, the opportunity to discuss its importance in depth is lost of the compiled information. But an important and very complete figure was inserted (Figure 1), which I hope meets your request and expectations.

Regarding the "Minor concerns" all were carried out and as can be seen the article was reviewed in its written, grammatical and scientific structure by MDPI itself.
Line 12: delete "It", change "for" into "by". (change made)
Line 20: "Clostridium" should be italic. (change made)
Line 22: change "tract" into "tracts". (change made)
Line 28:  "C. " should be italic. (change made)
Line 36: "Clostridium" should be italic, revised elsewhere needed. (change made)
Line 40: change "although" into "but". (change made)
Line 77: delete double blanks after "birds". (change made)

I'm really sorry if we weren't able to meet all your expectations, but I hope that the new version, with all the contributions, can demonstrate to you the quality, importance and relevance of this review.

Sincerely,
Felipe Masiero Salvarani

Reviewer 3 Report

Comments and Suggestions for Authors

The review provides an in-depth examination of Clostridium chauvoei, a bacterium associated with blackleg in cattle and gas gangrene in various animals, highlighting its impact on veterinary medicine. The genus Clostridium encompasses numerous bacterial species, with C. chauvoei identified as a major pathogen causing significant economic losses globally. The article explores the bacterium's characteristics, clinical manifestations, and the main virulence factors contributing to diseases like blackleg. These factors include various toxins such as alpha toxin, beta toxin (DNase), hyaluronidase, delta toxin, neuraminidase, and a recently identified toxin, CctA. The study also highlights the historical context of blackleg vaccination in Brazil and underscores the ongoing challenges despite widespread immunization efforts. It discusses the diversity of vaccination protocols and the evolving understanding of immune responses, emphasizing the importance of continuous research for optimizing vaccine formulations. The article concludes by emphasizing the significance of accurate diagnostic methods, including direct immunofluorescence, immunohistochemistry, and polymerase chain reaction, in identifying C. chauvoei infections.

The article is well-written, fluid, and highly comprehensive concerning the characterization of C. chauvoei. The topic is intriguing, particularly considering the impact of blackleg in Brazil. However, the section dealing with the description of the management of this pathology should be further developed. Given the significance of vaccination for blackleg control, it is essential to reference the currently most employed vaccine type, providing a more detailed explanation of its major issues. The bibliography in this specific paragraph is limited and somewhat outdated, and, in general, the entire article relies on a low number of citations that are not very up-to-date for a review. It would be interesting to delve into the role of the CctA toxin for immunization purposes, as several recent studies have focused on its potential application in this context.

Comments on the Quality of English Language

 Minor editing of English language required

Author Response

Dear Reviewer 3,
Thank you for recognizing the scientific quality of the work, when you mention "The article is well-written, fluid, and highly comprehensive concerning the characterization of C. chauvoei".

Regarding his comments about "vaccination for blackleg control" we understand that the available literature on the subject was addressed and that was what was available in the literature. And it is a generalist, as there is little information available about the vaccine, demonstrating the need for more studies focusing on the development of new immunobiologicals to control blackleg.

But we included a further discussion about the role of the CctA toxin for immunization according to your request, which I transcribe below.
"CctA is one of the toxins produced by Clostridium chauvoei, and it belongs to a family of toxins known as beta-pore-forming toxins. Recent studies have indeed explored the potential of using CctA as a component for immunization against blackleg disease. The idea behind this research is to utilize the toxin in a controlled manner to trigger an immune response in animals without causing the actual disease. By doing so, the animal's immune system can recognize and develop defenses against CctA, thereby providing protection against the bacterium that produces this toxin.These studies aim to develop vaccines that are effective, safe, and capable of providing robust immunity against Clostridium chauvoei infection without causing harm to the animals. However, while the research shows promise, it's important to note that vaccine development and testing involve various stages, including rigorous trials to ensure safety and efficacy before potential implementation in veterinary practices[79,80]."

Sincerely,
Felipe Masiero Salvarani

Reviewer 4 Report

Comments and Suggestions for Authors

1/ L12: The sentence is grammatically incorrect. and Blackleg is not a “highly contagious disease”;

2/ L24: “lethality” is the same as “mortality”;

3/ L50-52: The sentence is unclear;

4/ L59: clonic signals - ?

5/ L61: clonic paralysis - ?

6/ L63: similar to what?

7/ L70: The “parasite/host ratio” is not addressed in the work;

8/ L78: The phrase “close contact between these agents and domestic animals” is not clear;

9/ L81: Not only domestic animals! Deer can also be infected;

10/ L112-113: “mother-of-pearl buds” is not a term that has been used in microbiology. Does it refer to the surface, colour or consistency of colonies? “denture edges” - ?

11/ L116-119. This is obvious and refers to many groups of bacteria;

12/ L126: “Ingestion of spore-contaminated pasture” - ?;

13/ L130: “swollen lesions” - ?

14/L183: should be hyaluronidase?

15/ L192: propagation of what?

16/ L216-219: What is the size of sialidase of Cl. chauvoei – 135 or 150 kDa?

17/ L244: “in strains of isolates” - ?

18/ L245: the reference strain, not sample;

19/ L246: What pore-forming toxin is responsible for anthrax signs?

20/ L254-256: Should be: Roberts. The O-antigen is rather referred to a polysaccharide of Gram-negative bacteria. Please use a more recent reference to support this statement;

21/ L266: “second outer membrane” - ?

22/ L270-272: This is a well-known fact that could be omitted;

23/ L311-316: Who has wanted to determine the most immunogenic part of the Cl. chauvoei

 Cell – the authors of the present work or Tamura et al. (1984 and 1987)?

24/ L357-358: The sentence is not clear; “MT” refers to a vaccine and a bacterial strain; this must be differentiated;

25/ L377-383: Why anthrax? 

Comments on the Quality of English Language

Moderate editing of English language required

Author Response

Dear Reviewer 4,
Thank you very much for reviewing our article, thank you very much for all your suggestions and I apologize if it does not represent a significant contribution to the field for you. But we had a lot of work and a lot of scientific dedication to prepare these reviews and as reviewer 1 states, "The authors have conducted a review of blackleg, focusing on the agent and management of the disease. It is always very interesting to find a review where the authors summarize all the information available on a topic. In this case, the authors take into consideration all the necessary aspects that are related to Clostridium chauvoei such as virulence factors, prevention and control, and vaccine production and vaccination protocols. The manuscript is very complete, interesting, and well written". I agree that in your point of view the article may not be good now, but certainly with the corrections suggested by all three reviewers and also the extensive review of the English writing of the manuscript that will be done by MDPI itself, I strongly believe that we will have a I work much better.

Below I will list each of your suggestions and the modifications we made, all of your questions are excellent and much of the problem lies in the poor technical quality of writing in English, I'm sorry:
1). Correction and rewriting of the sentence. Indeed Blackleg is not a “highly contagious disease” but rather "Blackleg it causes for Clostridium chauvoei, is a bacterial disease that primarily affects cattle..."
2). Correction and rewriting of the sentence. "...a disease with high mortality."
3). Rewriting the sentence. "Clostridia are responsible for the production of 20% of the main and most potent toxins of bacteriological origin [3]." Here we want to reiterate that the genus Clostridium produces around 20% of the most potent bacteriological toxins studied, as stated in the literature, demonstrating the importance of the genus and the diseases caused by Clostridia.
4). Correction and rewriting of the sentence. "...specific clinical signs..."
5). Correction and rewriting of the sentence. "...spastic paralysis..."
6). Here we mean that different species of Clostridium, despite producing different toxins, end up causing diseases with similar clinical signs. Myonecroses, for example, can be caused by the following species alone or together: C. septicum, C. chauvoei, C. novyi type A, C. perfringens type A, and C. sordellii, but the lesions are similar regardless of the etiology of the disease. In this case, malignant edema or gas gangrene can be caused by one of all of the agents mentioned above and cause similar clinical signs.
7). Correction and rewriting of the sentence. "The present article aims to review the main virulence factors and prevention of C. chauvoei infection and provide a compilation of data on Brazilian studies of blackleg."
8). Correction and rewriting of the sentence. "...gas gangrene requires, for example, the contamination with these agents in wounds resulting from surgical practices and even vaccinations without aseptic care."
9). Correction and rewriting of the sentence. "Unlike gas gangrene, blackleg is an endogenous infection that affects domestic and wild ruminants,..."
10). Correction and rewriting of the sentence. "...including regular shapes or irregular colonies and a rough surface."
11). Yes, and that's why it applies to C. chauvoei. But if the reviewer thinks we should remove this sentence, we can do so. Just remembering that it is a literature review, and although it seems "This is obvious and refers to many groups of bacteria" it is important that new readers, such as undergraduate students, who do not know much about microbiology, can have access to this information.
12). Correction and rewriting of the sentence. "Ingestion of spores in feed, pasture and contaminated water is the main route of transmission..."
13). Correction and rewriting of the sentence. "...increase in the volume of affected muscles..."
14). Correction and rewriting of the sentence "...hyaluronidase..."
15). Correction and rewriting of the sentence. "...propagation of agents and toxins..."
16). Correction and rewriting of the sentence. "...150 KDa..."

17 and 18). Correction and rewriting of the sentence. "The gene encoding CctA is conserved, having been identified in strains off different continents, as well as in the reference strain of the ATCC (10092)."
19). Correction and rewriting of the sentence. It wasn't anthrax but blackleg. And the pore-forming toxin responsible for clinical signs is Toxin A (CctA).
20). Correction made, rewriting of the sentence and changing the reference by Mattar, M.I.; Cortiñas, T.I.; Guzmán, A.M.S. Immunogenic protein variations of Clostridium chauvoei cellular antigens associated with the culture growth phase. FEMS Imm. Med. Microb. 2002, 33, 9–14, https://doi.org/10.1111/j.1574-695X.2002.tb00565.x
21). Correction was made, the sentence was rewritten and the issue of "second outer membrane" was removed. In fact in the cited article the description was "The surface of the negatively stained cell walls appeared to be covered with indentations or perforated by many small holes. However, in ultra-thin section most of the cell walls had a distinct two-layered structure which showed little sign of perforation or indentation". But in our manuscript it was wrong. Thanks.
22). Correction made and sentence deleted.
23). Correction made, rewriting of the sentence. The conclusion belongs to Tamura 1984 and 1987 and not to us authors of this manuscript. Sorry.
24). In Brazil, the first vaccine studies were carried out based on studies with a strain called MT. And the vaccine developed from the MT strain was also called the MT vaccine. That's why the confusion and that's why I deleted the term MT vaccine strain.
25). Correction made, rewriting of the sentence. It's not anthrax, it's blackleg.

Sincerely,
Felipe Masiero Salravani

Round 2

Reviewer 2 Report

Comments and Suggestions for Authors

Although the authors addressed some issues in the previous version, the response to reviewer is poorly presented, which should be presented in a point-to-point response, but not a general and vague description. Meanwhile, some issues were not replied.

Comments on the Quality of English Language

Moderate editing of English language required

Author Response

Dear reviewer 2,
I'm sorry once again if I wasn't clear in my answers, but I tried to answer all your questions.

1). "The review lacks of logic, which is difficult to read. The authors can present the review in a more logical style, such as: Blackleg, Pathogen,
Pathogenicity, Diagnosis, Prevalence, Prevention and control." And I answered clearly and not in a general way, because despite thinking that "the review lacks of logic, which is difficult to read", we do not agree and we are sure that we approached a correct way the topics "Blackleg, Pathogen, Pathogenicity, Diagnosis, Prevalence, Prevention and control." Furthermore the article was evaluated by 4 reviewers and below I transcribe their speeches to be able to refute their comment "The review lacks of logic, which is difficult to read", perhaps the quality of the English could have caused this impression. But look at what feedback the article received: "The authors have conducted a review of blackleg, focusing on the agent and management of the disease. It is always very interesting to find a review where the authors summarize all the information available on a topic. In this case, the authors take into consideration all the necessary aspects that are related to Clostridium chauvoei such as virulence factors, prevention and control, and vaccine production and vaccination protocols. The manuscript is very complete, interesting, and well written." (Reviewer 1). "The review provides an in-depth examination of Clostridium chauvoei, a bacterium associated with blackleg in cattle and gas gangrene in various animals, highlighting its impact on veterinary medicine. The genus Clostridium encompasses numerous bacterial species, with C. chauvoei identified as a major pathogen causing significant economic losses globally. The article explores the bacterium's characteristics, clinical manifestations, and the main virulence factors contributing to diseases like blackleg. These factors include various toxins such as alpha toxin, beta toxin (DNase), hyaluronidase, delta toxin, neuraminidase, and a recently identified toxin, CctA. The study also highlights the historical context of blackleg vaccination in Brazil and underscores the ongoing challenges despite widespread immunization efforts. It discusses the diversity of vaccination protocols and the evolving understanding of immune responses, emphasizing the importance of continuous research for optimizing vaccine formulations. The article concludes by emphasizing the significance of accurate diagnostic methods, including direct immunofluorescence, immunohistochemistry, and polymerase chain reaction, in identifying C. chauvoei infections. The article is well-written, fluid, and highly comprehensive concerning the characterization of C. chauvoei." (Reviewer 3). Are we all wrong in seeing that the article is logical and easy to read? Unfortunately I don't have to restructure an entire article, based on the opinion of only one reviewer, when others have different opinions, including the authors.

2. "The review can be visualized by using suitable Figures and Tables to summarize main points". Another suggestion, at the discretion of us authors and other reviewers, is not necessary, since copying figures and tables from other articles would not add information, but would only make it more summarized. And as I had explained to you, Our objective was to do a real review without photos and tables, so that everyone could read the information. Several studies in the area of education and teaching demonstrate that review articles with photos and tables are not as effective in disseminating knowledge, since when summarizing information in a table or figure, the opportunity to discuss its importance in depth is lost of the compilation in Figures and Tables.  But we will insert a figure that I hope will contribute to the quality of the article. Figure 1

These two points were what you mentioned as Major concerns. And thank you for agreeing that "Although the addressed authors some issues in the previous version", but saying that "reviewer is poorly presented, which should be presented in a point-to-point response, but not a general and vague description. Meanwhile, some issues were not replied." We respond point-to-point and not to general and vague description. We tried to respond to you clearly and kindly, that your view on "Major concern" was respected, but that we authors, based on the other reviewers, maintained the structure of the review article as it was sent to you. Once again I apologize, but it was the most polite way we had to answer your questions.

In Minor concerns:

1. There are many spelling and formatting problems, e.g. (performed spelling).

Line 12: delete "It", change "for" into "by" (change made)

Line 20: "Clostridium" should be italic (change made)

Line 22: change "tract" into "tracts" (change made)

Line 28: "C. " should be italic (change made)

Line 36: "Clostridium" should be italic, revised elsewhere needed. (change made)

Line 40: change "although" into "but" (change made)

Line 77: delete double blanks after "birds" (change made)

2. The format of reference list needs to be revised, e.g., some references have doi no., but not for all. At this point, we were unable to insert all references with a DOI, as older articles do not have one, but out of 82 references, only seven articles do not have a DOI.

And regarding English, it was reviewed by the MDPI English service itself, but I have already contacted the magazine's editor, to express his still dissatisfaction with his comment "Moderate editing of English language required".

Sincerely,
Felipe Masiero Salvarani

Reviewer 4 Report

Comments and Suggestions for Authors

“Blackleg: a review of the agent and management of the disease in Brazil” (R1) – a review

Comments to the Authors:

Only a little progress was made when compared to the previous version and some concerns have not been addressed.  

1/ L91: The sentence needs some attention

2/ L102: The statement “macrophages play an important role in the pathogenesis of blackleg” is an oversimplification;

3/ L165-166: The colony morphology is still unclearly described (“regular shapes” -?);

4/ L170: The sentence needs some attention

5/ L178: The sentence needs some attention

6/ L260: The sentence needs some attention

7/ L275: similar in what?;

8/ L312: “pores with up 50 subunits” – is unclear

9/ L315: There is one or several types of neuraminidase/sialidase produced by Cl. chauvoei? phrase “close contact between these agents and domestic animals” is not clear;

10/ L322: “hydrolysis of erythrocytes” - ?

11/ L335: Erythrocytes have no cell wall.

12/ L487-490: a/ The work of Mattar et al. (53) is not cited in the text; b/ That article does not mention “O-antigen”; c/ Again, use a more recent reference than that of Henderson (1932 – sic!) to support the statement of O-antigen in  Cl. chauvoei;

13/ L644: what antibodies ?

14/L685-688: The sentence is still unclear; the Authors have only changed “sample” to “strain”;

15/ L788: “lethality” is the same as “mortality”;

16/ L788: “the rapid course of the disease is not fully understood” - ?

Comments on the Quality of English Language

Author Response

Dear reviewer 4,
Once again, we thank you for your new review, pointing out more errors so that we can improve the quality of the article. And I inform you that it was reviewed by MDPI English Service to avoid problems with the English language.

Responding in detail to your new comments point by point:
1/ L91: The sentence needs some attention. Corrected sentence ("..for example, the contamination with these agents in wounds resulting from surgical practices without aseptic care.")

2/ L102: The statement “macrophages play an important role in the pathogenesis of blackleg” is an oversimplification; The sentence was redone to not be so simplified. ("In summary, macrophages are integral to the immune response against Clostridium chauvoei in blackleg disease. While they play a critical role in attempting to eliminate the bacteria and their toxins, their actions can contribute to both the resolution of the infection and the development of tissue damage due to the inflammatory response. Understanding the interplay between macrophages, the immune response, and the bacteria is crucial for developing strategies to modulate the immune system effectively, potentially reducing the severity of blackleg and improving treatment outcomes.")

3/ L165-166: The colony morphology is still unclearly described (“regular shapes” -?); Sentence redone to be clearer for the reader. ("...colonies regular with low convex, whitish-grey, large (5 mm in diameter), but may also form a continuous "carpet" of bacteria, (the bacteria are swarming) or as apleomorphic in structure").

4/ L170: The sentence needs some attention. Stence redone. ("Therefore, it is necessary to use immunological diagnostic methods [25] such as direct immunofluorescence [26,27] and PCR [13,28] to identify C. chauvoei.")

5/ L178: The sentence needs some attention. Sentence redone. ("The primary transmission route involves ingesting spores present in contaminated feed, pasture, and water...")

6/ L260: The sentence needs some attention. Sentence redone. ("Hence, Gram-positive bacteria capable of producing hyaluronidase can initiate infections in the mucosa, subcutaneous tissue, and/or skin...")

7/ L275: similar in what?; Seventeen redone. ("...similar mechanisms of action of..."

8/ L312: “pores with up to 50 subunits” – is unclear. Sentence corrected.

9/ L315: There is one or several types of neuraminidase/sialidase produced by Cl. chauvoei? One type ("Likewise, C. chauvoei is known to produce one neuraminidases that play a significant role in the pathogenesis of the infection...")

10/ L322: “hydrolysis of erythrocytes” - ? Sentence corrected. ("...destruction of erythrocytes...")

11/ L335: Erythrocytes have no cell wall. The expression walls reiterated.

12/ L487-490: a/ The work of Mattar et al. (53) is not cited in the text; b/ That article does not mention “O-antigen”; c/ Again, use a more recent reference than that of Henderson (1932 – sic!) to support the statement of O-antigen in Cl. chauvoei; The reference by Mattar et al. [53] was inserted into the text and a more up-to-date reference was added Ziech et a. [17] 2018.

13/ L644: what antibodies ? Sentence redone. We are talking about vaccine antibodies against cell wall of the flagellum of C. chauvoei.

14/L685-688: The sentence is still unclear; the Authors have only changed “sample” to “strain”; Sentence modified to try to make it clearer. ("In light of these findings and the continued occurrence of blackleg in vaccinated herds, studies have conducted comparisons between the MT vaccine and local samples. Results from guinea pigs vaccinated with commercial products and subsequently exposed to the MT strain revealed that 95.46% of the tested vaccines failed to provide protective immunity.")

15/ L788: “lethality” is the same as “mortality”; No. Lethality: Refers to the proportion of cases of a disease that result in death. For example, if a disease affects 100 animals and 10 of them die, the lethality would be 10%. Mortality: It is the measure of the total number of deaths in relation to the population at risk during a given period. For example, if in a population of 1000 animals, 100 die from a specific disease in a year, the mortality rate would be 10%. Therefore, lethality focuses on the proportion of cases of a disease that are fatal, while mortality refers to the rate of deaths in relation to the population at risk.In this case the correct term is lethality.

16/ L788: “the rapid course of the disease is not fully understood” - ? Modified sentence for "acute course of the disease"

Once again, I thank you for the enormous opportunity to allow us to improve our article with your pertinent contributions. I hope we were able to meet your expectations with the new article.

Sincerely,
Felipe Masiero Salvarani
